# Ecumenism and Communism in the Romanian Context: Fr. Andre Scrima in the Archives of the Securitate

**Iuliu-Marius Morariu**

Faculty of Orthodox Theology, Babes-Bolyai University, 400692 Cluj-Napoca, Romania; iuliu.morariu@ubbcluj.ro

**Abstract:** Based on information offered by the archives of the former Romanian Securitate, this paper presents the way in which the image and actions of Fr. Andre Scrima, one of the most important Romanian theologians from abroad (and the representative of the Ecumenical Patriarch Athenagoras to the Second Vatican Council), were a topic of interest for the aforementioned surveillance institution during the Romanian Communist regime. At the same time, the emphasis was on the relevance of the ecumenical area for the regime of Bucharest and its interest in using it in order to create the illusion of democracy and freedom. Ninety-five years after the birth of the Romanian theologian, who was not only the disciple of the Romanian Patriarch Justinian and of the aforementioned Greek Patriarch but also an important writer, professor, and abbot, this article aims to bring attention to his life and work, and to provide an investigation into an aspect of his biography which has not been investigated by contemporary researchers from the fields of theology and history until now.

**Keywords:** ecumenism; Romania; Orthodoxy; Securitate; Romanian Orthodox Patriarchate





## 1. Introduction

An important personality both for the Romanian context and for the ecumenical landscape, Fr. Andre Scrima (1925–2000) is an author often mentioned by theologians from different backgrounds. His works are quoted, his manuscripts are published (see for example: Scrima 1996, 2003, 2004, 2005a, 2005b, 2007, 2008a, 2008b, 2018), and his ideas are brought into debate (Dumitrașcu 2016, pp. 272–81; Tofan 2019; Grigorean 2009, pp. 69–82; Manolescu 2015, pp. 33–42; Morariu 2020, pp. 497–511). During his lifetime, he was also a personality with important contributions, which caused important personalities of his time to contact him, exchange letters, or speak about his work (Puyo 1975, pp. 144–48).

In the Romanian context in which the Securitate tried to control everything, due to his international visibility and impact, he also became important to this institution. Aware of this aspect and of the fact that ecumenism was among the important topics for the aforementioned structure, we examined how Fr. Scrima's image was reflected in the archives of the Securitate. We emphasise the main aspects that caused the aforementioned institution to be interested in his activity and work, and what information it collected about him. We not only present the main aspects that defined his image for the surveillance agents but present the way in which ecumenism is reflected in these documents. Due to the fact that the dossiers kept in the aforementioned archives were written between 1950 and 1966, we refer only to this period. This does not mean that the Romanian surveillance institution was not interested in him and his work after this time but, most likely, the documents reflecting it have been lost.

One may ask why such information is relevant. It must be said that although Fr. Scrima's work has been published and analysed in different situations and contexts, his Securitate dossier was not previously published. This is precisely the new aspect of this article. By making available this unpublished information, we approach the subject of Fr. Scrima's image and emphasise both how the Romanian surveillance bodies were interested in his work and how this influenced his activity in the ecumenical field.

## 2. Ecumenism and Communism in the Romanian Context: Fr. Andre Scrima in the Archives of the Securitate

### 2.1. An Overview of Scrima's Activity

Having left Romania in 1956 with a scholarship offered by the Indian government to complete a PhD in Benares, Andre Scrima did not only receive this degree but also another PhD in Paris and he became a professor, an abbot, and the delegate of the Ecumenical Patriarch Athenagoras (Gheorghiu 2009) to the Second Vatican Council after refusing to return to Communist Romania. This made him a real target for the Securitate agents who followed him, his ideas, attitudes, and works, and tried to intercept his correspondence with Romanian friends[1] and family.

Clearly, we must also note that he was followed by the Securitate agents for various reasons even before leaving Romania.[2] Therefore, from 16 March 1953, the "Vântu" source offered long and detailed notes about him, his activity, and the situation of his family (ACNSAS 1952, p. 110). There were also earlier notes regarding his life and activity. For example, on 24 July 1950, the "Vlaicu" source informed the Securitate archives about the reorganisation of the Antim Monastery in Bucharest and the structure suggested by Sofian Boghiu, its abbot:

> "The priest SOFIAN BOGHIU, the abbot of Antim monastery in the Capital, proposed and the priest accepted that the following be recommended in positions of responsibility at the Antim monastery:
>
> -ANDREI SCRIMA secretary.
>
> -Rev. NEOVIL REABCU cashier
>
> Rev. NEOVIL is a Bessarabian, who lived in a monastery with the priest SOFIAN BOGHIU-Dobruşa-Soroca Monastery.
>
> In the refuge, he was ordained at the Putna-Rădăuţi monastery. Here, at Antim, he works as a tailor." (ACNSAS 1952, p. 216)

The document reveals both how well-informed the aforementioned institution was and how interested it was in the religious life of monasteries. The information was most probably provided by somebody from within the system, an acquaintance of Fr. Boghiu or Scrima himself. Later, in 1953, the same source "Vântul", together with another one called "Nicolau", offered regular information about him, Valeriu Anania, and others of his friends (ACNSAS 1952, pp. 195–206; Morariu 2020, p. 500). One of the most important reasons why he was followed and accused by the Securitate was the fact that he joined the "Burning bush" (Plămădeală 2002; Chiulescu 2017; Ciornea 2015; Diaconescu 2018), a mystical movement about which he also wrote in his books (Scrima 1996) in which he testified about the influence of its leader, John Kulâghin, in his life (Dumitraşcu 2016, p. 274). On 11 March 1954, the sources of the surveillance institution were informed about one of Scrima's meetings taking place at Antim (ACNSAS 1952, p. 192). They were also informed about his relationships with important personalities at the head of the Orthodox Church with a far-right orientation, such as the Metropolitan Firmilian Marin of Oltenia[3], and about his interactions with people from abroad, theologians or otherwise. One example includes his correspondence with Professor Habib who helped him obtain the scholarship in Benares (ACNSAS 1952, p. 293). Their correspondence was also intercepted by state organs. Therefore, in the second volume of the surveillance dossier, there was a letter from 22 August 1955, translated from English and addressed both to Patriarch Justinian and to Fr. Andrei. It stated the following:

> "Badar Bach 22th of August 1955
>
> To His Holiness Justinian,
>
> Bucharest
>
> To father Andrei

Dear brother Andrei,

After I met you for the second time and decided to come on a pilgrimage to Budha's country, I took the necessary steps, but no action was taken—I spoke to the Minister of Education and he also acknowledged that it would be fair to invite here a student from Romania and to be awarded a scholarship of 200 Rupees per month—these scholarships are only given upon recommendation by the ambassadors.

I addressed our ambassador in Yugoslavia, who is also the ambassador for R. P. R. You must absolutely seek to see Dr. R. Dayal, the ambassador, coming to Bucharest from time to time; but he does not have a permanent office there. Ask him and he will decide in this regard. Compliments to the patriarch and the other priests.

Habib." (ACNSAS 1952, p. 293)

### 2.2. His Relationships with the Romanian Patriarchate

The fact that the Romanian authorities were informed about Fr. Scrima's intention to leave the country to study in Benares is therefore obvious. However, for reasons that we cannot fully explain, he was allowed to leave the country and arrived in Benares after stopping for a while at the Bossey Ecumenical Institute in France and on Month Athos. The unscheduled stopovers and his refusal to return led the Romanian surveillance institution to make inquiries in order to discover who allowed his departure. Mr. Iliescu, the employee of the Ministry of Culture who helped him with the passport, was removed and asked to write a declaration on this subject (ACNSAS 1952, pp. 77–79). The "Costea" source was among the ones who were asked to write what he knew about him and this subject. Shortly after his departure, he wrote the following:

"Patriarch Justinian, who, for many years, used Andrei Scrima as a personal adviser and kept him in the patriarchal palace for days, filling him with various favours and attentions, tried to obtain the necessary passport. The Patriarch pleaded that, in India, there were Orthodox Romanians, of whom no one had been in charge until then; Andrei Scrima left the country in the autumn of 1956. References about his civil attitude were picked from Mr. Iliescu, who, at that time, held an executive position within the Ministry of Cults; I gave unfavourable references, so that, after a few days, I was summoned to the Ministry of Cults by com. Iliescu; he was later retired on age grounds and admonished, given that others, who knew him better, described him very favourably". (ACNSAS 1952, pp. 53–54)

This note, together with a similar one, led the Securitate to interrogate his former friends, including Valeriu Anania (Anania 2017, pp. 57–59), to intercept his correspondence with Patriarch Justinian and try to see who the Patriarch's intermediaries were in his dialogue with Scrima.[4] In this context, a letter from 29 September 1957 was added to his surveillance dossier, which was written by the Romanian monk from Benares (ACNSAS 1952, pp. 314–15), in which he mentioned that he arrived well, was accepted to doctoral studies, and needed a scholarship from the Patriarch in order to continue his studies until the documents requested by the Indian state were ready. The request was accompanied by an acknowledgement released by the Benares Hindu University (ACNSAS 1952, p. 316), in which his status was mentioned. Shortly afterwards, another letter followed, through which he addressed a formal request to the Patriarch (ACNSAS 1952, p. 317) in order to be able to receive an answer for his request from the previous letter. From Patriarch Justinian, the Securitate kept only a small telegram sent to Fr. Scrima on 17 November 1960, in which he asked to visit Metropolitan Justin Moisescu in order to receive news from him (ACNSAS 1952, p. 279). Patriarch Justinian wrote him about Fr. Moraru, most probably in order to inform the Romanian monk about the fact that the Securitate knew about their correspondence and he refused to say things that could be used against him.

This letter, together with other documents from the Communist period (Ciuceanu and Păiușan 2001, p. 18), prove that the Romanian leader of the Orthodox Church between 1948 and 1976 knew how to maintain good relationships with the state, giving the impression of cooperating, while at the same time protecting the interests of the Church.

This is all the more important in this context and Fr. Scrima's interviews often spoke about Justinian and his activity under persecution. Such an example includes the interview he gave to Olivier Clément shortly after his arrival to France, reviewed in the "Ortodoxia" journal by Valeriu Anania,[5] which aroused the security informers' interest. Under these circumstances, shortly after its publication, the organs of state requested different translations of the text and summaries of its context. This was the moment when the Romanian Securitate became more interested in ecumenism, while other Romanian exiled writers such as Virgil Gheorghiu (Ghillyboeuf 2019; Gheorghiu 1999, 2002; Morariu 2018, pp. 149–55) started to denounce the Communist maltreatment of the Romanian Church. The aforementioned "Costea" source underlined the following:

> The article written by Andrei Scrima and published by Olivier Clément in French is entitled "The Romanian Orthodox Church or the miracle of the incessant prayer." It is a kind of interview, sprinkled with quotations from the words of Patriarch Justinian whose content is, in short, the following.

> -The Romanian Orthodox Church has kept, under Communism, a relatively privileged and average situation of superiority compared to the Russian Church during the Soviet Union.

> -Within the Romanian Patriarchate, the network of ecclesiastical schools has remained almost intact (ten seminaries and two higher theological institutes) and now depends only on the Church, which pays special attention to it.

> -The Romanian Orthodox Church currently has five publishing houses and regularly publishes three patriarchal magazines and five metropolitan magazines, all able to compete with the best theological journals of the west.

> -The Church has recently (?) received the right to teach high school religious education in state schools and operates somewhat under a regime approved by the state, which helps in terms of material necessities.

> -This privileged situation is explained by the exceptional personality of Patriarch Justinian, an old friend of Gheorghe Gheorghiu Dej, whom he hid from the police (?) during the repression of a strike.

> -Patriarch Justinian, perfectly loyal to the state, has placed the activity of the Church above any policy from the beginning:

> -"In the thinking that guides the cultural and economic life of the present Romanian state, none of the ideas that constitute the overall vision of any religion and even more of the Christian religion of Orthodox confession, can be found," and yet, each remaining on his positions and maintaining the hope of a peaceful triumph, "an atmosphere of peace and respect characterises the relationship between the state and the church"[6] (quoted from a speech by Patriarch Justinian)." (ACNSAS 1952, p. 55)

*2.3. Scrima's View on International Matters Reflected in the Archives of the Securitate*

The lack of a real and objective evaluation of the situation of the Romanian Orthodox Church made the Securitate angry with Fr. Andrei Scrima and they began to carefully watch whether he corresponded with others and what books and journals the Romanian bishop received from his disciple (ACNSAS 1952, p. 94); other aspects, however, caused the Securitate to become anger. The first aspect concerned the fame that he started to have in the west. Little by little, he became one of the most respected Orthodox theologians from the west as it can be seen from what Catholics such as fr. Yves Congar say about his work (Puyo 1975, pp. 144–48) and the Romanian organs of repression were aware of this

aspect. The second aspect, in connection with the first, concerned the fact that he was not controllable from Bucharest and was known as someone who was against communism. His attitudes and actions displayed at the conferences he was invited to speak at, in important places such as UNESCO[7] or in different universities and religious spaces, during which he militated for the union of the Churches and tried to speak about the relevance of a dialogue within the Orthodox space, also made him known as a vertical man, who disapproved of communism and often referred to the persecution of the Romanian clergy on religious grounds. In other cases, he tried to defend his friends who remained in this space from the persecution of the regime.[8] As a consequence, the Securitate sent people to watch him carefully and even interrogate him, and, when possible, those who were in contact with him. Among them, besides various Catholic or Lutheran priests, there were also lay people. One example concerns the architect Gheorghe Baramki with whom Scrima had a few meetings with in the 1960s when he visited Israel on occasion. Later on, he was also questioned by the Romanian officials. A note from 1964 presents the following aspects regarding their meetings, which demonstrate how interested the Romanian political police were in the Romanian Archimandrite's activity and attitude:

> "The Romanian fugitive monk SCRIMA ANDREI works as a representative of the Patriarchate of Constantinople near Vatican. He resides in Rome in the Greek Orthodox parish of this city.

> As the representative of the Patriarchate of Constantinople, SCRIMA ANDREI has the task of conducting negotiations for the union of the Orthodox and the Catholic churches.

> In January 1964, SCRIMA ANDREI visited Jordan, where, through the architect BARAMKI GHEORGHE, he showed his interested in the activity of the Romanian Orthodox Church, as well as in the situation of the priests ANANIA, FELIX, SOFIAN, BENEDICT, among others, who had been arrested after his fleeing the country.

> As BARAMKI GHEORGHE was going to visit Romania in July 1964, they established that SCRIMA would come to Jordan again before leaving BARANKI to give him some indications and then to find out the result of the visit.

> Not being able to travel, SCRIMA ANDREI asked BARAMKI for details of the visit in a letter sent in October 1964." (ACNSAS 1952, p. 4)

It was not only his relationships with people from Israel that interested the organs and informers of the Securitate. His attitude towards the Israeli context was also among the aspects that raised their interest. They were therefore informed about his attitude against Jewish people and regarding the fact that he supported the Arab cause, which placed him in a certain conflict with Jewish people and with some local Christians.[9] At the same time, the informers knew about his relationships with the Vatican, with Constantinople, and about his teaching activity. Thus, a note belonging to the "Radu" source, dating from the beginning of the eighth decade of the 20th century, tried to summarise the information regarding his activity, bringing attention to the following aspects:

> "By 1962, he moved to Istanbul and won favour with the ecumenical patriarch of Constantinople, who, after a while, sent him as his representative to Vatican. From Scrima's accounts and from some Catholic prelates, who were the representatives of Vatican in that place, it appears that he has had a major contribution to the improvement of relations between Vatican and the Ecumenical Patriarchate of Constantinople, along the lines of eliminating the schism between Catholicism and Orthodoxy. In Vatican, A. Scrima has built a series of relationships that support him as a professor in Lebanon. A. Scrima performs a prodigious philosophical activity in favour of Vatican and against the mosaic religion." (ACNSAS 1952, p. 88)

Therefore, his activity as a writer, professor, and priest in the ecumenical area were topics of interest for the Romanian surveillance services. The fact that he became a representative voice for the Orthodox area increased their curiosity but this attitude was not new, as we have already tried to demonstrate. Like in the case of other people who refused to return to Romania (for example, Chesarie Gheorghescu (Gheorghescu 2009)), the authorities tried to persuade him to come back but without any results. Not only were his writings, interviews, and speeches at conferences monitored but also his sermons. In the beginning of the seventh decade of the last century, specifically on 25 November 1962, a source from Istanbul informed the authorities in Bucharest of the fact that he had made some allegations against the communist regime. The report underlined the following:

> "We are informed from Istanbul that, on 25 November 1962, a certain Andrei Scrima, archimandrite, who fled the country, during a religious service, officiated at the Balik-Pazer Church in Istanbul, made a series of slanderous statements with respect to our country.
>
> The residence should notify them if the item is known to the Central body and, if necessary, it should remove the house which is undesirable to the metropolitan authorities." (ACNSAS 1952, p. 132)

It was not only Fr. Scrima who was under the surveillance of state institutions but also those who were in contact with him or corresponded with him, as we have already mentioned. What is interesting is that not only Valeriu Anania, Benedict Ghiuş, or Patriarch Justinian were investigated because of their correspondence with the Romanian Archimandrite, but also those who were listening to him or reading his articles in the Romanian area. The investigation of documents reveals an interesting situation dating from 8 January 1966, when Petre Vasilescu was imprisoned and interrogated because he listened to Fr. Scrima's programme on Radio Free Europe and spoke in his group of friends about the way in which he presented the lifting of the recent anathemas in the relationship between the Catholic and Orthodox churches (ACNSAS 1952, p. 121).

### 3. Conclusions

As we have tried to emphasise in our article, Fr. Andrei Scrima was an important personality of the Orthodox space who, because of his fame and activity in the ecumenical area, was constantly under the surveillance of authorities.[10] His attitude against the communist regime and in favour of an ecumenical space were carefully monitored by them and his activity against the dictatorial regime of Bucharest made authorities keep a close eye on him. His friends who corresponded with him and those who listened to him on the fadio or read his articles were often not only under supervision but also were imprisoned, cross-examined, and forced to denigrate him. The people from abroad who contacted him (such as architect Baramki) were also questioned and intercepted, while his articles were translated and analysed by the authorities. One of the most important aspects to be mentioned is his correspondence and relationship with Patriarch Justinian, who was his mentor and who never abandoned him.

On a final note, we would like to underline the fact that, based on the documents kept in the archives of the Securitate in Bucharest, the activity of Fr. Andrei Scrima in the ecumenical space represented an important subject for the Romanian dictatorial regime, which is why the regime carefully monitored him, his family, his friends, and disciples, without using them as factors of pressure.

**Funding:** This research received no external funding.

**Informed Consent Statement:** Not applicable.

**Data Availability Statement:** Not applicable.

**Conflicts of Interest:** The author declares no conflict of interest.

## Notes

1     For example, in 1951, they tried to make him inform state organs about Anania and his activity but he refused to do so; retrieved from the Archives of the National Council for the Study of the Securitate (hereinafter referred to as ACNSAS 1952, p. 201).

2     Shortly before his departure, when he requested a derogation in order to take his exams earlier, the sources also informed the Securitate about this and about the way in which he was invited to study in India. "Andrei Scrima" (the monastery brother who has not yet received his tonsure) makes requests to the Theological Institute to take the equivalence exams rapidly and without too much preparation in order to obtain the degree in theology as soon as possible. It is known that he had a degree in philosophy and was an assistant in the old Faculty of Philosophy but had not yet obtained his degree in theology. He had the Patriarch's approval to take, without frequency and internship exemption, all theology and BA exams. So far, Scrima had not rushed but then began rushing. Fârtăţescu (chief executive) said that "soon Scrima will go to India, sent by the Patriarch to study". Fârtăţescu added: "Last year, an Indian dignitary came to visit the patriarchal palace and through his intercession Scrima arranged to receive an official invitation to move to India as soon as possible. Now Scrima has received that invitation and the patriarch is preparing his departure. Before he leaves, Scrima also wants to finish with theology exams." (ACNSAS 1952, p. 160).

3     "Andrei Scrima said he would go to spend the Chris tmas holidays with Firmilian. Firmilian was still in Bucharest (he lived in the patriarchal palace). Fencing said he would leave for Craiova with the Metropolitan. Dec. 18 1953 Nicolau." (ACNSAS 1952, p. 193). On 16 June 1955, the "Cornel" source also informed the Securitate that he had close relationships with Firmilian Marin with whom he often talked to through the whole night and that often bishop Antim Nica often gave him a lift with his car." (ACNSAS 1952, p. 177).

4     Therefore, on 10 September 1958, the Securitate was informed that Marina asked Glicherie Moraru, a priest from the United States, to offer the USD 600 scholarship requested by Fr. Scrima. (ACNSAS 1952, p. 131). For more information about Fr. Moraru and his activity, see also: (Anania 2008; Gârdan 2007, 2010, pp. 790–820).

5     A fact already known by the "Costea source" who wrote about it shortly after its publication and led the Securitate to make inquiries: "What is sure is that Patriarch Justinian helped him with money (sometimes sent through others) during his travels, and with preparing his departure, according to some of his confessions, a series of theological and material manuscripts, and documentaries (this is the result, among other things, of the report published by Andrei Scrima, under the signature of Olivier Clément, in the weekly protesting "Reforms", which was published in Paris, no. 644 of 20 July 1957, a report commented afterwards by Bart[olomeu] Anania, with praises and interpretations, in an issue of the journal "Orthodoxy", edited by the Romanian Patriarchate.)" (ACNSAS 1952, p. 54).

6     In fact, as it is already known, at that time, the situation of the Romanian Orthodox Church under the communist regime was rather complicated and the institution had to face many restrictions and interventions of the regime in its activities.

7     For example, in 1969, the following was mentioned in one note by the same "Costea" source: "Last year he was invited by UNESCO to hold a conference on ecumenical issues." (ACNSAS 1952, p. 52).

8     Even in the articles dedicated to subjects such as the Philocalic collection, in which he underlined the crucial role played by Fr. Dumitru Stăniloae in their translation, in an attempt to transform him into a personality with international visibility and therefore to make the communist regime think twice before harshly persecuting the Romanian theologian who had been just imprisoned. See: (Scrima 1958a, pp. 295–398; 1958b, pp. 493–16; 1958c, pp. 293–94; ACNSAS 1952, pp. 107–8).

9     They were, for example, informed about the cultural reactions that concerned him. A source wrote the following at the end of the 1960s: "Shzurakiss wrote a book entitled "Letter to a Christian Friend," indirectly addressed to Scrima." (ACNSAS 1952, p. 89).

10    They therefore knew even how he looked and had his precise description. Therefore, in an article from 6 April 1960, the following was mentioned: "ANDREI SCRIMA, a monk, aged 31–32, was sent 3–4 years ago to India to study this side of mysticism or to another faculty (I don't know exactly). Fit, with an aquiline nose and brown hair. Clever, he knew French, English and some German. A theologian, originally from the Slatina monastery-Moldova, where he was a novice. Quiet and moral in all aspects. Later on, a librarian at the Patriarchal headquarters, from where he left to India, having even a special invitation from President Nehru, when he came to the country. He took part in the reception provided and served as a translator for the issue of the Patriarch and his departure to India. After he left, he wrote a few letters and then he didn't write anymore. As long as he was in the country, he had had no quarrel with anyone, and had made no other intrigues, neither big, nor small. I know nothing more about him than what I showed he did in Lebanon. His current domicile is said to be in France or Switzerland." (ACNSAS 1952, p. 47).

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
