# Peer review of "Ecumenism and Communism in the Romanian Context: Fr. Andre Scrima in the Archives of the Securitate"

_religions, doi:10.3390/rel12090719_

Round 1

Reviewer 1 Report

The article provides with the synthetic presentation of the Orthodox theologian. It is an interesting contribution for the historians of the ecumenical movement, mainly because of using historical sources, like the documents of Securitate. Yet, I suggest making a profound revision of the article's structure, namely to divide the main chapter (chapter 2) into two or three parts. Moreover, I suggest carefully proofreading the text, perhaps by the native speaker, as there are many grammatical and stylistic errors. 

Author Response

Many thanks to the reviewer. I hav segmented the main chapter, as it can be seen and the text has been proofreaded (I can also attach if needed a document where it can be seen how the proofreader changed certain aspects).

Reviewer 2 Report

The article has the potential of being published, but not in its actual form.

The author aims to present some of the intelligence gathered by Securitatea about Scrima between 1950 and 1966.  He or she seems to work with the assumption that because Scrima was an important theologian whatever Securitatea wrote about him must be important. This is partially true, but what Securitatea wrote about Scrima is relevant for an academic audience, only if it brings to light some aspects of Scrima's life or personality that were not previously known. This does not appear to be the case here. The author should explain why he or she slected that particular material from Scrima's Securitatea file and why he or she stopped at 1966. Where there no other files about Scrima after that? Finally, the article needs extensive English editing. The style and the language are often difficult to follow and at time one needs to guess what the author is trying to convey. 

Author Response

Thank you. I have explained why the information stop in 1966, why it is an important theologian and which is the novelty aspect of the investigation. In the same time, the text have been proofreaded. I also attach a version when it can be seen how the proofreader changed different aspects.

Round 2

Reviewer 1 Report

The author introduced revisions suggested in the previous review. 

Author Response

Many thanks to the reviewer!

Reviewer 2 Report

The author explained why he or she examined only the particular period from Scrima’s life, but other areas of concern remain.

First and foremost, the author has not explained why the information he or she has found in the Securitatea file is of any significance. What does it bring new to the existing literature?

Second, the article has not been proofread properly. A slight improvement can be felt, but the text still abounds in English language mistakes. To give just some random examples: lines 79-80 the verb informed is used with the preposition about when it should be used with the preposition on; line 120, it is not clear what is meant by, "his civil attitude"; line 168-170 speak about theological magazines (most probably theological journals) who can compete with "the best theological police (??) in the West". Also, Scrima's first name should be written either as in Romanian Andrei or as in French André, but not Andre.  

Third, some statements should be clarified, others referenced. Line 187 begs the question: what was the real and objective situation of the Romanian Orthodox Church at the time? At line 192, the author affirms that Scrima became "one of the most respected Orthodox theologians in the West". He or she should mention by whom and put a reference to the statement.

Author Response

Many thanks to the reviewer.

Regarding the significance, I have explained better and marked in red color why it is important. Regarding the proofread, the text was now proofreaded by a scholar who is  native speaker. I have also completed the suggested aspects, as it can be seen from the text. Regarding  the name, I let the editor to decide which version wishes. 

I have also explained  what was the real and objective situation of the Romanian Orthodox Church at the time? And also put a reference for the line 192. 

With gratitude,

the author

Round 3

Reviewer 2 Report

I appreciate that the author has followed most of the suggestion received so far. The article improved visibly and its relevance for the field became clearer. Nonetheless, despite the assurance received from the author, the English remained virtually unchanged. To give just one example. In the introduction to the second version one reads at lines 38-40: “Of course, this does not mean that the Romanian surveillance institution was not interested in him and his work after this year but, must likely, the documents reflecting it were lost.” The expression there should have been most likely and not must likely. The same mistake is repeated in the current version. If the editor takes responsibility for a proper proofreading of the text, the article can be published.

This manuscript is a resubmission of an earlier submission. The following is a list of the peer review reports and author responses from that submission.